# The Contribution of the Zebrafish Model to the Understanding of Polycomb Repression in Vertebrates

**DOI:** 10.3390/ijms24032322

**Published:** 2023-01-24

**Authors:** Mariette Hanot, Ludivine Raby, Pamela Völkel, Xuefen Le Bourhis, Pierre-Olivier Angrand

**Affiliations:** Univ. Lille, CNRS, Inserm, CHU Lille, UMR9020-U1277–CANTHER–Cancer Heterogeneity Plasticity and Resistance to Therapies, F-59000 Lille, France

**Keywords:** zebrafish, epigenetics, PRC1, PRC2, morpholino, gene editing

## Abstract

Polycomb group (PcG) proteins are highly conserved proteins assembled into two major types of complexes, PRC1 and PRC2, involved in the epigenetic silencing of a wide range of gene expression programs regulating cell fate and tissue development. The crucial role of PRC1 and PRC2 in the fundamental cellular processes and their involvement in human pathologies such as cancer attracted intense attention over the last few decades. Here, we review recent advancements regarding PRC1 and PRC2 function using the zebrafish model. We point out that the unique characteristics of the zebrafish model provide an exceptional opportunity to increase our knowledge of the role of the PRC1 and PRC2 complexes in tissue development, in the maintenance of organ integrity and in pathology.

## 1. Introduction

Polycomb group (PcG) genes were originally discovered in *Drosophila melanogaster*, where they were shown to be involved in the maintenance of homeotic (*Hox*) gene repression and body plan specification [1,2]. PcG genes were subsequently found to be conserved in vertebrates, including humans [3]. Extensive work over the past two decades has demonstrated that PcG proteins are not only involved in the silencing of *Hox* gene expression but indeed regulate a wide variety of gene expression programs controlling a plethora of cellular and developmental processes [4,5,6]. Furthermore, alterations in PcG gene function have been implicated in a number of human diseases, including cancer [7,8,9,10,11].

PcG proteins are typically organized as two main multiprotein complexes involved in transcriptional repression through the post-translational modification of histones. The Polycomb repressive complex 1 (PRC1) monoubiquitylates histone H2A at Lysine 119 (H2AK119ub1) [12,13], whereas the Polycomb repressive complex 2 (PRC2) catalyzes mono-, di- and trimethylation of Histone H3 at Lysine 27 (H3K27me1/2/3) [14,15,16]. PRC1 and PRC2 are recruited at the same genomic sites to form Polycomb chromatin domains where H2AK119ub1 and H3K27me3 epigenetic marks are enriched and gene expression is locally precluded [4,5,6,17,18].

Importantly, systematic biochemical characterization of Polycomb repressive complexes in *Drosophila* and mammalian cells has demonstrated that PRC1 and PRC2 are both organized as sets of diverse assemblies through the association with distinct auxiliary proteins [8,19,20]. Although the association with various accessory subunits does not change the histone-modifying activities of the two Polycomb repressive complexes, the auxiliary proteins play a role in the recruitment of the complexes to chromatin, their stabilization and/or the modulation of their activity.

The mammalian PRC2 catalytic activity relies on the association of the histone lysine methyltransferase EZH2 (or its paralog EZH1) with the subunits EED, SUZ12 and RBBP4/7 [15]. In addition, this PRC2 core complex interacts with various auxiliary subunits to define at least two distinct PRC2 complexes: PRC2.1 and PRC2.2 [21,22,23,24] (Figure 1A). PRC2.1 includes one of the three PCL proteins (PHF1, PHF19 or MTF2), together with EPOP, LCOR (PAL1) or LCORL (PAL2), while the auxiliary subunits of PRC2.2 are JARID2 and AEBP2. The auxiliary factors regulate PRC2.1 and PRC2.2 H3K27 methyltransferase activities in several ways: by increasing PRC2 binding to chromatin (e.g., PHF1, MTF2, JARID2, AEBP2), by interacting with histone post-translational modifications (e.g., PHF1, PHF19, JARID2) and/or by the regulation of the allosteric activation of the PRC2 core complex (e.g., PHF1/19, MTF2, LCOR/LCORL, JARID2). To complicate matters further, it has been reported that EZH2 could control gene expression in a PRC2-independent fashion [25].

The PRC1 core complex is composed of the E3 ubiquitin ligase RNF2 (or its paralog RING1) that catalyzes H2AK119ub1, associated in dimer with one of the six Polycomb group RING finger (PCGF) proteins. The binding of the PCGF protein stimulates the E3 ligase activity of RING1/RNF2 on one hand and imposes the choice of the auxiliary proteins incorporated into a given PRC1 assembly on the other hand. PRC1 complexes are usually classified as either canonical PRC1 (cPRC1) or non-canonical PRC1 (ncPRC1) [26,27,28]. In addition to the RING1/RNF2, cPRC1 complexes assemble PCGF2 or BMI1 (PCGF4) together with one of the chromobox proteins (CBX2, CBX4, CBX6, CBX7 and CBX8) that bind H3K27me3, one of the polyhomeotic proteins (PHC1/2/3) essential for Polycomb-mediated repression and eventually an SCM protein (SCMH1 or SCML1/2/3). In contrast, ncPRC1 can be assembled around all six PCGF proteins and include RYBP or its paralog YAF2, as well as various additional auxiliary proteins depending on the PCGF subunit present in the complex forming distinct ncPRC1 assemblies such as ncPRC1.1, ncPRC1.2/4, ncPRC1.3/5 and ncPRC1.6 (Figure 1B). It is well established that RYBP/YAF2 significantly stimulates the E3 ubiquitin ligase activity of RING1/RNF2 [29,30,31], but the function of the other auxiliary subunits is far less understood. However, some of these accessory proteins might be involved in the recruitment of ncPRC1 to chromatin. For instance, the H3K39me2 demethylase KDM2 contributes to the targeting of ncPRC1.1 at unmethylated CpG islands, independent of its demethylase activity [32,33,34]. Similarly, the ncPRC1.6 auxiliary subunits MGA-MAX recognize the E-box DNA motif, whereas E2F6 heterodimerizes with DP-1 to bind to E2F recognition sequences [35,36,37]. Thus, a fundamental difference between cPRC1 and ncPRC1 relies on whether their recruitment to chromatin depends on PRC2-deposited H3K27me3. cPRC1 complexes are recruited to H3K27me3 via the recognition of the marks by the CBX2, 4, 6–7 proteins, whereas ncPRC1 assemblies are recruited to chromatin by their auxiliary subunits, independently of H3K27me3 (Figure 2).

Most of the studies aiming at the understanding of Polycomb repression are genetic and biochemical analyses performed in *Drosophila*, mouse and embryonic stem cells. More recently, the zebrafish model emerged as an additional system shedding light on the role of Polycomb repression in vertebrate development. The zebrafish (*Danio rerio*) is a small freshwater teleost (bony fish) native to Southeast Asia [38]. It emerged as a model for studying early development in the 1950s but was quickly used in broader research fields [39]. The success of zebrafish in research is due to its characteristics that make it an exceptional experimental model. In particular, (i) zebrafish are robust and easy to maintain at low husbandry costs; (ii) zebrafish have a high fecundity rate, producing around 100–200 embryos per clutch and per week; (iii) zebrafish embryos develop externally and are optically clear; (iv) the zebrafish genome is fully sequenced, and comparison to the human reference genome revealed that about 70% of human genes have a zebrafish ortholog [40]; (v) thousands of zebrafish mutants coming from large scale mutagenesis screens are available [41]; and (vi) a variety of genetic engineering approaches, including transgenesis, morpholino-mediated gene expression knockdown and genome-editing technologies, can be applied to the zebrafish model [42,43,44].

Here, we describe the zebrafish genes coding for the PcG proteins and review the genetic data addressing the function of the PcG genes during zebrafish development. Additionally, we discuss how the zebrafish model offers new insights into the study of Polycomb repression in vertebrates.

## 2. The Zebrafish Genes Coding for the Components of the PRC1 and PRC2 Complexes

During evolution, a whole-genome duplication, known as the teleost genome duplication (TGD), occurred in the teleost lineage before the expansion of bony fish species [45,46]. Ohnologs produced by the TGD were presumably redundant, and a copy was subsequently lost randomly [47]. Nevertheless, around 15% to 20% of the genes remained duplicated in teleost fishes after the gene loss events [48]. Table 1 illustrates the status of genes encoding the subunits of the PRC1 and PRC2 complexes in the zebrafish genome.

A search for zebrafish orthologs of the PRC1 and PRC2 subunits revealed that 13 components out of 48 exist as pairs of ohnologs in the zebrafish genome. The genes retained duplicated code for subunits of the core PRC2 complex (*suz12a, suz12b*), auxiliary proteins of PRC2 (*jarid2a, jarid2b*), subunits of the core PRC1 complex (e.g., *bmi1a, bmi1b*), auxiliary proteins of cPRC1 (e.g., *phc2a, phc2b*) and auxiliary proteins of ncPRC1 (e.g., *rybpa, rybpb*).

Surprisingly, a number of genes, notably those coding for some PRC1 core subunits are absent in the zebrafish genome [49] (Table 1). These genes, likely lost during the rediploidization process responsible for a massive loss of duplicated genes having occurred after the TGD, include the catalytic subunit RING1 as well as the Polycomb group RING finger proteins PCGF2 and PCGF3. Hence, in zebrafish, PRC1-mediated monoubiquitination at H2A119K is solely accomplished by Rnf2. Loss of the *RING1* ortholog is also observed in other fishes, but it is not a general feature in teleost species (Table 2). For instance, the Japanese pufferfish (*Tetraodon nigroviridis*) has lost *ring1*, while the gene is retained in the medaka (*Oryzia latipes*) genome.

In the zebrafish genome, the Polycomb group RING finger protein genes *pcgf1* and *pcgf3* are present as singletons and *bmi1a/b* and *pcgf5a/b* are present as pairs, but *pcgf2* and *pcgf3* are both absent. Loss of *pcgf2* and *pcgf3* might not affect the diversity of the PRC1 complexes since the zebrafish c/ncPRC1.2/4 and ncPRC1.3/5 complexes would be assembled around Bmi1a/b and Pcgf5a/b, respectively. As for RING1/RNF2, the status of the PCGF orthologs is variable from fish to fish. In medaka, *bmi1*, *pcgf2*, *pcgf3*, *pcgf5* and *pcgf6* are all present as singletons but *pcgf1* is absent, whereas in the Japanese pufferfish, *pcgf3* is the only gene lost (Table 2). This contrasts with the H3K27me3-binding component of the cPRC1 assemblies. Indeed, none of the Cbx subunits has been lost during fish evolution; rather, *cbx* genes are often retained as pairs in most fish species, including zebrafish [50] (Table 3). It is not known whether the strict maintenance of all paralogs, as well as many ohnologs, in fish reflects the absence of functional redundancy between the Cbx family members and contributes to a higher complexity of cPRC1 assemblies involved in the remarkable diversity of the teleost fishes or whether it is a consequence of a selection pressure applied to neighboring genes in the blocks of synteny.

The other genes encoding PRC components not found in zebrafish include the RBBP7 subunit of the PRC2 core complex and HDAC2, which is an auxiliary protein of ncPRC1.6. Retinoblastoma-binding proteins 4 and 7 (RBBP4 and RBBP7) are highly similar, sharing about 90% amino acid identity. These proteins are members of the WD40 repeat protein family folding into a β-propeller allowing simultaneous binding to the H3 and H4 histone dimer and the PRC2 complex [51]. In addition, the individual knockdown of *Rbbp4* or *Rbbp7* does not impair embryonic development, while the simultaneous knockdown of the two genes causes embryonic lethality during the morula-to-blastocyst transition [52]. To date, there is no known functional difference between RBBP4 and RBBP7 within PRC2, and consequently, the absence of RBBP7 orthologs in zebrafish is unlikely to affect PRC2 function. Histone deacetylases HDAC1 and HDAC2 are auxiliary proteins found in the ncPRC1.6 complex [26,53] coupling H2A119ub1 ubiquitination to the deacetylation of histones H3 and H4. Because the inactivation of both *Hdac1* and *Hdac2* is required to generate a phenotype in mouse tissues, it is believed that the activity of HDAC1 and HDAC2 is mostly redundant [54,55]. Moreover, individual *Hdac1/2* disruption in embryonic stem cells (ESCs) is not associated with changes in chromatin modifications performed by the enzymes, such as histone H3 acetylation (H3ac) or H3K27ac [53,56]. Thus, the absence of an HDAC2 ortholog in zebrafish is not expected to change the conservation of ncPRC1.6 function or the H2A119ub1–H3/H4 deacetylation coupling.

Hence, an investigation into the zebrafish genes encoding PRC1 and PRC2 subunits reveals that in spite of the loss of several genes coding for PRC components, the diversity and complexity of the complexes are conserved in this organism. This makes the zebrafish a good model for studying Polycomb repression during development.

## 3. Genetic Approaches to Study Zebrafish Polycomb Group Gene Function

During the last two decades, the development of powerful reverse genetic approaches has propelled the investigation of gene function in zebrafish.

Morpholinos (morpholino phosphorodiamidate anti-sense oligonucleotides; MOs) are oligonucleotide analogs using morpholine rings to replace the ribose backbone [57]. MOs are resistant to nucleases and act through a stable base pairing with RNA interfering with mRNA splicing or protein synthesis and resulting in a penetrant gene knockdown in zebrafish [58]. Although MO-mediated gene knockdown cannot be considered a real genetic manipulation because genomic DNA remains unchanged, MOs have proven to be a powerful tool for the assessment of gene function during zebrafish development. MOs, like other gene knockdown-based approaches, have a number of limitations. In particular, since MOs do not induce heritable genetic alterations, the degree of knockdown could be variable and the efficacy is limited to about 3 days post-injection in zebrafish embryos. Another drawback relies on potential off-target and toxicity effects. In particular, MOs tend to activate the Tp53 pathway leading to non-specific phenotypes of embryonic defects and neuronal apoptosis, although these effects could be partially limited through the co-injection of tp53-MOs [59]. In addition, as the generation of genetic mutations has become more important in zebrafish genetics, a number of studies pointed out discrepancies between MO-mediated and knockout-mediated phenotypes [60,61,62,63], even if in a number of cases, these differences could be due to a genetic compensation mechanism in the mutants [64]. Nevertheless, MO-based knockdown remains a precious tool for a rapid assessment of gene function during zebrafish development.

Targeting induced local lesions in genomes (TILLING) was the first reverse genetic method applied in zebrafish in order to identify point mutations generated through N-ethyl-N-nitrosourea (ENU) chemical mutagenesis, in a defined gene [65]. The approach has been applied at a large scale [41], and tens of thousands zebrafish mutants are available. However, in spite of being extremely powerful, TILLING suffers from several limitations. The method relies on random chemical mutagenesis, and it might statistically be more difficult to obtain mutants for small-size genes; some sequences could have a weak mutagenic potential, mutations in a given gene might not be located at a suitable position, and several other mutations might be present in the zebrafish genome in addition to the mutation of interest.

The implementation of the programmable site-specific endonucleases, as the most efficient and versatile tool to manipulate any genomic sequence, revolutionized reverse genetics in zebrafish and bypassed the limitations of the previous technologies [66,67]. These programmable site-specific endonucleases, the zinc-finger nucleases (ZFNs), the transcription activator-like effector nucleases (TALENs) and the clustered regularly interspaced short palindromic repeat (CRISPR) RNA-guided Cas9 nucleases (CRISPR/Cas9) allow a precise modification of the zebrafish genome at any gene and at a chosen position.

These different reverse genetic tools have been applied to study the molecular function of a number of genes coding for PRC1 and PRC2 components in zebrafish (Table 4).

Complementary to gene knockdown and knockout experiments, pharmacological approaches to inactivate PRC1 and PRC2 function could be applied. PRC complexes have been reported to be involved in the growth of various tumors and are thus considered as targets for cancer therapy. Therefore, extensive searches for molecules that would inhibit PRC enzymatic activity or stability have been conducted. Several potent EZH2-specific inhibitors, including GSK126 [101], have been developed. These inhibitors bind to the catalytic domain of EZH2 and compete with S-adenosyl-methionine, the methyl donor of the methyltransferase activity. In addition to EZH2 catalytic inhibitors, other small molecules target the H3K27me3 binding pocket of EED and inhibit the allosteric activation of EZH2 by preventing the EED–H3K23me3 interactions [102,103]. Another approach to inhibit PRC2 activity is based on the disruption of the interaction between EED and EZH2 using peptide inhibitors, the drug astemizole or the natural compound wedelolactone [104,105,106]. Several PRC1 inhibitors have also been reported. In particular, RB-3 directly binds to RNF2-BMI1, blocks the association of RNF2-BMI1 with chromatin and inhibits H2AK119 ubiquitylation [107]. However, to date, only few studies have applied pharmacological approaches to study Polycomb repression in zebrafish [68,73,108].

## 4. Polycomb Group Proteins Support the Development and Viability of Zebrafish

### 4.1. Role of PRC1 in Zebrafish Development

Most of the experiments addressing the function of PcG genes in vivo have been conducted in mice, and both cPRC1 and ncPRC1 have important roles in mouse development [8]. Homozygous and heterozygous *Ring1* mutant mice are viable but present anterior transformations and other abnormalities of the axial skeleton [109], whereas *Rnf2* is essential for mouse embryogenesis as *Rnf2* loss of function results in lethality during gastrulation [110]. This difference indicates that Ring1 and Rnf2 have non-redundant functions during mouse development. The zebrafish has a single gene, *rnf2*, coding for the catalytic subunit of cPRC1/ncPRC1 complexes. Surprisingly, ZFN-mediated inactivation of *rnf2* generates embryonically viable zebrafish mutants allowing the study of *rnf2* in vertebrate development [80]. The mutants die at around 4–5 days post-fertilization (dpf) and exhibit developmental defects. In particular, *rnf2* mutants lack the pectoral fins, likely due to failure in the terminal differentiation dependent on Fgf signaling as the fin developmental program initiates properly with the expression of *tbx5*. In addition, cranial neural crest-derived cartilage precursors migrate into the pharyngeal arches but fail to differentiate into chondrocytes, leading to severe craniofacial defects in *rfn2* zebrafish mutants at about 72 h post-fertilization (hpf) [111]. The reason why *rnf2*-deficient zebrafish mutants survive gastrulation, while *Rnf2* mutant mice do not, remains unclear. One possible explanation is that the maternally deposited Rnf2 mRNA and protein are sufficient to allow early embryonic development. Indeed, in zebrafish, maternal products are expected to be degraded in the maternal-to-zygotic transition (MZT) which takes place at around cleavage cycle 10, whereas MZT occurs at cleavage cycle 1 in mice [112]. In this line, it has been shown that *Ring1* and *Rnf2* heterozygous embryos generated from double *Ring1*- and *Rnf2*-deficient oocytes and wild-type spermatozoids cannot develop after the 2twocell stage, demonstrating the essential contribution of the Ring1/Rnf2 maternal products in the very early developmental stages [113].

From the Polycomb group RING finger genes, knockout mice have been generated for *Pcgf2* [114], *Bmi1* [115], *Pcgf3* and *Pcgf5* [116], and *Pcgf6* [117], whereas *pcgf1* is the only PCGF member for which a knockout zebrafish line has been described [83]. TALEN-mediated inactivation of *pcgf1* does not affect the viability nor the fertility of zebrafish. However, the growth rate in early developmental stages is reduced in the absence of *pcgf1* gene function, and a significant number of mutant fish show signs of premature aging. Since pcgf1-deficient fish are viable until adulthood, it might be deduced that ncPRC1.1 is dispensable in zebrafish. In this regard, it is worth noting that the *pcgf1* gene is absent from the genomes of the Tetraodontidae *Tetraodon nigroviridis* and *Takifugu rubripes* [49]. Alternatively, a genetic compensation mechanism could rescue the loss of ncPRC1.1 in the *pcgf1*-deficient fishes. Such a genetic compensation mechanism in the mutant zebrafish line could also explain the marked MO-pcgf1 phenotype [82]. MO-pcgf1 embryos have a small head, a reduced or even absent telencephalon and a reduced body size. MO-mediated *pcgf1* knockdown is responsible for an increase in the expression of neural marker genes such as *sox2*, *otx2* and *ngn1* leading to an abnormal activation of neural induction, together with a decrease in the expression of the pluripotent markers *oct4*, *hes1* and *nanog*. This abnormal neural induction and inhibition of neural stem cell self-renewal might then account for the morphant phenotype.

So far, the investigation of the individual contribution of each cPRC1 and ncPRC1 variant complex in the development remains challenging, in part because only a small number of mutants have been described, but also because PRC1 auxiliary proteins could be shared with cellular processes unrelated to the Polycomb repression. In zebrafish, Rnf2 is the catalytic subunit common to all cPRC1 and ncPRC1 complexes, and its loss of function leads to the death of embryos at 4–5 dpf after the implementation of the body plan [80]. In contrast, Yaf2 is a component of the ncPRC1 variant complexes but is absent in cPRC1 assemblies. Moreover, Yaf2 is redundant in ncPRC1 since it could be replaced by one of its two paralogs Rybpa and Rybpb. However, MO-*yaf2* injection into embryos arrests zebrafish development before somatogenesis at around 16 hpf [88]. The fact that the phenotype of *yaf2* morphants is much stronger than the phenotype of the *rnf2* mutants indicates that Yaf2 might have an ncPRC1-independent function in zebrafish development, in addition to its contribution to the ncPRC1 activity, and complicates the study of the function of the distinct PRC1 complexes. A comparable picture has been described in mice. Pcgf6 and L3mbtl are two components of the ncPRC1.6 complex and while *Pcgf6* mutants survive to adulthood [117], *l3mbtl* mutants display a strong embryonic phenotype and die at around the time of gastrulation [118].

### 4.2. Role of PRC2 in Zebrafish Development

PRC2 plays a crucial role in development as depletion of the core subunits Ezh2, Eed and Suz12 in mice is embryonic lethal around gastrulation [119,120,121]. In zebrafish, mutants are also available for *ezh2*, *eed* and *suz12a/b* (Table 4), but in contrast to mice, mutants deficient for PRC2 function gastrulate and form their body plan normally but die between 8 and 12 dpf [72,73,75,76]. The presence of the maternally deposited products may explain the correct development of the zygotic mutants. The involvement of maternal PRC2 has been explored through the exposure of zebrafish embryos at the 1–2 cell stage to Ezh1/2 inhibitors [73,108,122]. Pharmacological inhibition of maternal PRC2 elicits additional developmental defects, but embryos survive gastrulation and develop, indicating that maternal PRC2 is not required for early development and global implementation of the body plan. Moreover, the generation of maternal zygotic (MZ) *ezh2* mutant embryos through germ cell transplantation revealed that *MZezh2* mutants gastrulate and form a normal body organization but die at around 2 dpf [72]. Thus, maternal PRC2 contributes to zebrafish development but is not required for the very early steps.

Rbbp4 and Ezh1 are two other subunits of the PRC2 core complex. Zebrafish mutants lacking *rbbp4* gene function are lethal between 5 and 10 dpf and show a severe neurogenic phenotype with microcephaly and microphthalmia [78,123]. However, this phenotype is not necessarily fully assigned to *rbbp4* loss of function since Rbbp4 has also PRC2-independent actions on chromatin metabolism, such as actions on histone deacetylation and chromatin assembly. In contrast to all other PRC2 core complex components, *ezh1*-deficient zebrafish mutants are viable, fertile and without obvious phenotype, indicating that Ezh1 is dispensable for zebrafish development [69], as is also the case in mice [124,125].

EZH1 and EZH2 are both able to monomethylate H3K27 in vitro, but the role of PRC2 in the deposition of H3K27me1 in vivo still remains a point of discussion [126,127,128,129]. Knockout of *Eed* and *Suz12* in mouse ESC lines as well as PRC2 knockdown experiments in ESCs result in a global loss of H3K27me2 and H3K27me3 levels, but at best in a partial reduction of H3K27me1 [121,130,131]. The study of bulk histone modifications in *eed*-deficient zebrafish larvae at 9 dpf revealed a dramatic decrease in H3K27me2 and H3K27me3 marks, but unchanged global levels of H3K27me1, suggesting that monomethylation of H3K27 is independent of the PRC2 activity in the mutants [75]. Because EHMT1 (Glp) and EHMT2 (G9a) have been shown to be able to monomethylate H3K27, both in vitro and in mouse ESCs [132,133,134], their zebrafish orthologs Ehmt1a, Ehmt1b and Ehmt2 could be the histone methyltransferases responsible for the maintenance of H3K27me1 levels in the absence of PRC2 activity.

Investigations on histone modifications in TALEN-mediated *eed* zebrafish mutants revealed that global H2AK119ub levels are still maintained in the absence of PRC2 function at 9 dpf [75], presumably as a consequence of the H3K27me3-independent recruitment of ncPRC1 complexes. This parallels observations made in mouse ESCs where Ezh1/Esh2 double knockout does not impair H2AK119ub levels [127]. However, in another study using maternal–zygotic *MZezh2* mutants, it has been reported that loss of Ezh2 function from both maternal and zygotic origins results in the absence of H3K27me3 marks together with a dramatic loss of Rnf2 (cPRC1 and ncPRC1) recruitment to chromatin [135]. This puzzling study did not document whether H2AK119ub marks are also lost and contrasts with data from mouse ESCs where ncPRC1 has been shown to be recruited to chromatin independently of PRC2 [29,33]. One possible explanation could be that in *eed* mutants, the maternal PRC2 activity is sufficient to deposit enough H3K27me3 marks to initiate the recruitment of cPRC1 complexes in early developmental stages. Then, as development progresses, H2AK119ub modifications are propagated in a PRC2-independent manner through the action of ncPRC1 assemblies. Indeed, it has been shown that ncPRC1 could be recruited to chromatin through the binding of RYBP/YAF2 to H2A119ub marks. This hypothesis would explain the absence of Rnf2 recruitment to chromatin in *MZezh2* mutants and would explain the more severe phenotype of the *rnf2* and *MZezh2* mutants in comparison to *eed* and *ezh2* mutants. This is also in agreement with the order of Polycomb complex recruitment in early zebrafish embryos [136]. Before zygotic genome activation (ZGA), H3K27me3 marks are low–absent while H2AK119ub is present in chromatin. Then, at the ZGA stage, H2AK119ub enables PRC2.2 recruitment and H3K27me3 deposition.

Altogether, it is remarkable that zebrafish harboring mutation in genes coding for PcG genes and deficient for PRC1 and PRC2 activities survive gastrulation and die after body plan formation. This feature makes the zebrafish a particularly suitable model for studying the role of PRC1 and PRC2 in the formation and maintenance of various organs.

### 4.3. Function of Polycomb Repression in Zebrafish Digestive Tissues

A striking phenotype of the PRC2-deficient mutants is observed at the level of the intestine [73,75,137]. At 9 dpf, both *ezh2* and *eed* mutants present a marked alteration of the intestinal wall, which is strongly reduced and lacks folds at the level of the bulb. The intestine defects might prevent food uptake and might account for the larval death at about 12 dpf since that moment corresponds to the death time of unfed zebrafish larvae [138]. Interestingly, at 5 dpf, there is no structural difference in the intestinal wall between *ezh2*- or *eed*-deficient mutants and wild-type fish, suggesting that loss of PRC2 function does not impair intestine development but PRC2 activity is required for intestine maintenance. Histological analysis of the liver also shows defects in the organs of *ezh2*- or *eed*-deficient mutants at around 10 dpf [73,75,137]. In particular, the liver of PRC2-deficient larvae is characterized by a smaller size, disorganization in the organ and signs of steatosis associated with an increase in lipids and macrovesicles in the liver. In addition, a delay in pancreas development is also observed in PRC2-deficient larvae [73,75]. Intestines, livers and pancreases of zebrafish larvae lacking PRC2 function are smaller and disorganized but still express the tissue-specific terminal differentiation markers *fabp2* (fatty acid-binding protein 2, intestinal), *fabp10a* (fatty acid-binding protein 10a, liver basic) and *prss1* (serine protease 1, trypsin), respectively. This indicates that terminal differentiation of the digestive organs in unaffected in the absence of zygotic PRC2 function, yet the differentiation tissues are not maintained over time in the mutants. At the level of the digestive organs, the phenotypes of *ezh2*- and *eed*-deficient larvae are undistinguishable [73,75], excluding that Ezh1 and/or PRC2-independent Ezh2 functions could play a critical role in the development of these organs, at least until 12 dpf.

The *rnf2* mutation leads to a pleiotropic phenotype in 3 dpf zebrafish embryos, including mobility defects, lack of pectoral fins, defects in craniofacial development and pronounced heart edemas [80]. Moreover, in situ hybridization on *rnf2*-deficient embryos at 3 dpf using the terminal differentiation markers *fabp2*, *fabp10a* and *prss1* shows that loss of PRC1 function is responsible for a smaller intestine, an absence of terminal intestinal differentiation in the liver and the absence of the pancreatic lobe [139]. Thus, both PRC1 and PRC2 are required to maintain the differentiation status of digestive organs during zebrafish development.

### 4.4. Polycomb Repression in Cardiac Cell Identity and Function

*MZezh2* mutants develop a stringy heart as one of the most prominent phenotypes [72]. Morphologically, at 2 dpf, the heart of *MZezh2* mutant embryos fails to undergo cardiac looping and develops as a straight heart tube with a smaller ventricle. In situ hybridization analyses studying the expression of cardiac markers during the development coupled with time-lapse imaging on transgenic lines expressing GFP in the developing heart revealed that myocardial precursors are specified in the absence of maternal and zygotic Ezh2 function in early stages. In the mutant, at 1.5 dpf, the bending of the heart tube does not occur and some myocardial cells detach from the ventricle and the atrium to become dispersed over the regular heart tube over time. At 2 dpf, a partial loss of expression of the terminal differentiation marker *nppa* is observed in the mutants. Thus, in *MZezh2* mutants, it is likely that myocardial cells fail to maintain their cardiac identity, leading to structural instability of the heart [72].

Zygotic *rnf2* mutant zebrafish have also been reported to harbor similar defects in cardiac development [81,139]. At 3 dpf, the heart of *rnf2* mutants shows a tubular-shaped morphology. Using single-heart RNA-Seq analyses, Chrispijn et al. [139] showed that *tbx2/3* were upregulated in the heart of *rnf2*-deficient mutant at 2 and more importantly at 3 dpf. This loss of *tbx* gene repression could then be responsible for a reduction in chamber-specific gene expression, a misbalance in cardiac cell types and the absence of cardiac looping observed in PRC1-deficient embryos. However, PRC1 may also be involved in regulating cardiac contraction at no later than 24 hpf [81]. Indeed, *rnf2* deficiency disorganizes the sarcomere assembly in the zebrafish heart and causes defects in the conduction system [81].

### 4.5. Polycomb Repression in Zebrafish Developmental Hematopoiesis

To identify chromatin factors involved during developmental hematopoiesis, a large-scale MO-based screen targeting zebrafish orthologs of 425 human chromatin factors has been conducted [140]. This screen identified factors that affect the development of primitive erythroid progenitors or regulate the development of definitive hematopoietic stem and progenitor cells. These chromatin regulators involved in hematopoiesis are associated not only with chromatin remodeling complexes, acetyltransferase complexes or deacetylase complexes, but also with the Polycomb repressive complexes PRC1 and PRC2 [140]. The role of PRC2 in hematopoiesis was further demonstrated by showing that *ezh1* knockdown induces an increase in phenotypic hematopoietic stem and progenitor cells because of enhanced hemogenic endothelium commitment, occurring at the expense of arterial endothelium maintenance in the ventral wall of the dorsal aorta [68]. Moreover, in spite of an increased number of hemogenic endothelium cells transitioning to arterial-fated endothelium cells due to *ezh1* loss of expression, Ezh2 activity is subsequently required to increase hematopoietic stem and progenitor cell formation. This indicates that Ezh1 and Ezh2 do not play redundant or antagonistic roles in hemogenic endothelium specification/hematopoietic stem and progenitor cell formation but rather act as sequential regulators [68].

### 4.6. Polycomb Repression and Nervous System Development in Zebrafish

Whole-mount in situ hybridization experiments performed on *eed*-deficient larvae at 5 dpf revealed that the expression of some stemness, neuronal and glial markers is altered, suggesting that neuronal differentiation is impaired in the absence of functional PRC2 [75]. Interestingly, despite a ubiquitous expression of *eed* in the zebrafish brain, loss of *eed* function results in neuronal gene expression alterations in relatively discrete areas of the mutant brains. For instance, the expression of *neurod1*, a neuronal precursor marker, is specifically lost in a subset region of the hindbrain of *eed* mutants. Thus, the effects of the loss of PRC2 activity appear to be strictly cell-specific, context-dependent and differentially affecting cell fates [75]. PRC1 has also been shown to be involved in the development of the zebrafish neural system [82,141]. In particular, loss of *rnf2* function leads to abnormal migration and differentiation of neural crest cells and neural precursors [141]. The formation of 5-HT serotonin neurons and myelinating glial cells is also defective in *rnf2*-deficient zebrafish embryos [141]. In addition, ectopic expression of enteric nervous system markers is found in the forebrain of *rnf2* mutant embryos as well as in the retina of *ezh2* and *eed* mutant embryos [73,75,141]. Thus, both PRC1 and PRC2 are involved in the development of the enteric nervous and central nervous systems. Although the role of PcG proteins in nervous system development is not well studied, the zebrafish appears to be a suitable model for conducting these investigations.

### 4.7. Polycomb Repression in Cancer

PcG proteins control many essential cellular and developmental programs, and the alteration of genes coding for subunits of PRC1 and PRC2 complexes has been largely reported to be involved in tumorigenesis [8,11,142]. Although the zebrafish has been proven to be an excellent model for studying cancer development [143,144,145], the involvement of PRC1 and PRC2 in cancer biology still remains largely underexplored in zebrafish. However, a relationship between PRC2 and cancer development has been clearly established [76]. Homozygous loss of *suz12a* and *suz12b* functions is lethal at around 10 dpf, but the loss of three *suz12* alleles out of four (*suz12a^-/-^* and *suz12b^+/-^,* or *suz12a^+/-^* and *suz12b^-/-^*) significantly accelerates the onset and increases the penetrance of malignant peripheral nerve sheath tumors (MPNSTs) in a *tp53-* and *nf1*-deficient zebrafish model of MPNSTs. In addition, Suz12 deficiency expands the spectrum of tumor types in *tp53/nf1*-deficient fish, as lymphoid leukemia, soft tissue sarcoma and pancreatic adenocarcinoma are found in addition to MPNSTs. Interestingly, these tumors are also identified in the human spectrum of *SUZ12*-deficient malignancies indexed in the AACR Project Genomics Evidence Neoplasia Information Exchange (GENIE) database, thus validating zebrafish as a relevant genetic model for studying the involvement of PcG genes in cancer [76].

A link between PcG genes and zebrafish cancers was further established with the study of the zebrafish *asxl1* mutant [146,147]. *ASXL1* is one of the three human orthologs of the *Drosophila Asx* gene, which is highly conserved across various species [148]. ASXL1 is a regulatory subunit of the Polycomb repressive-deubiquitinase (PR-DUB) complex that counteracts the PRC1 action through the removal of the H2AK119ub mark [149,150]. The *ASXL1* gene has been found altered in several malignant myeloid diseases such as myeloproliferative neoplasms, myelodysplastic syndromes, chronic myelomonocytic leukemia and de novo or secondary cases of acute myeloid leukemia [151,152,153,154,155,156]. The TALEN-mediated homozygous inactivation of the zebrafish *asxl1* gene results in larval death by 14 dpf, with defects in the liver and the intestine and muscular atrophy [146]. By contrast, heterozygous *asxl1^+/-^* fish developed normally and were undistinguishable from their wild-type siblings but died prematurely starting at about 18 weeks of age. In addition, about half of the heterozygous *asxl1^+/-^* fish developed myeloproliferative neoplasms at 5 months, supporting the use of zebrafish as a model for studying the role of Polycomb repression in cancer.

### 4.8. Polycomb Repression and Behavior

The study of the locomotor activity of *eed*-deficient zebrafish larvae at 5 dpf during alternating light and dark phases revealed that the mutants are hyperactive compared to wild-type siblings. The hyperactive phenotype of PRC2-deficient mutants is particularly marked and significant during the dark cycles [75]. Interestingly, despite their hyperactive phenotype, the *eed* mutants do not present an increase in their exploration behavior in thigmotaxis assays. This observation shows that Polycomb repression and PRC2 function are involved in the control of zebrafish behavior, although the underlying neurological mechanism is still unknown.

Another insight into the involvement of the PRC2 complex in zebrafish behavior came from the study of the topoisomerase IIα (Top2a) [157]. Inhibitors of Top2a activity such as sodium salicylate and MO-*top2a*, as well as mutations in the *can4* gene encoding Top2a, alter the robust social preference for age-matched conspecifics at 3 weeks of age. In addition, RNA-Seq analyses performed on *can4*-deficient fish at 3 dpf show that Top2a depletion in zebrafish regulates autism risk genes, while ChIP-Seq data suggest that PRC2 and H3K27me3 mediate Top2a-dependent gene regulation. The antagonistic relationship between Top2a and H3K27me3 was further demonstrated by the use of specific Ezh2 inhibitors. Thus, Top2a plays a crucial role in promoting the development of social behavior in zebrafish by the maintenance of a regulatory network that selectively controls the expression of a large subset of autism risk genes. Furthermore, Top2a likely functions by antagonizing PRC2 function and H3K27me3-mediated gene silencing [157].

## 5. Concluding Remarks and Perspectives

PRC1 and PRC2 are families of chromatin-modifying complexes, which have been the focus of intense attention because of their fundamental role in the control of gene expression programs that govern a large variety of cellular and developmental processes. Research on the function of PRC1 and PRC2 in vivo largely relies on studies in *Drosophila* and mice. However, the zebrafish has recently emerged as a powerful genetic model for investigating many vertebrate physiological and pathological processes. In this review, we discuss the study of the role of the PRC1 and PRC2 complexes from a zebrafish perspective. The efficacy of reverse genetic tools allows the generation of various zebrafish models harboring knockdown or knockout genes encoding subunits of the PRC1 and PRC2 complexes (Table A1). Strikingly, and in contrast to mice, zebrafish deficient for PRC1 or PRC2 activities survive gastrulation and develop until the late embryonic or early larval stages, respectively. The implementation of the body plan together with the relative transparency of the zebrafish embryos allows an exceptional visualization of the development of the organs in the absence of PRC function. Hence, the zebrafish model allows light to be shed on the role of PRC1 and PRC2 in the maintenance of the differentiation state of the heart and the digestive organs, as well as in hematopoiesis and nervous system development. Furthermore, zebrafish mutants allow researchers to address the question of the involvement of PRC complexes in human pathologies, such as tumorigenesis, and generate models to study cancer development in humans. Going forward, research on Polycomb repressive complexes using the zebrafish model might hold great promise and present remarkable opportunities to unravel the function of PRC1 and PRC2 in various physiological and pathological processes in the coming years.

## Figures and Tables

**Figure 1 ijms-24-02322-f001:**
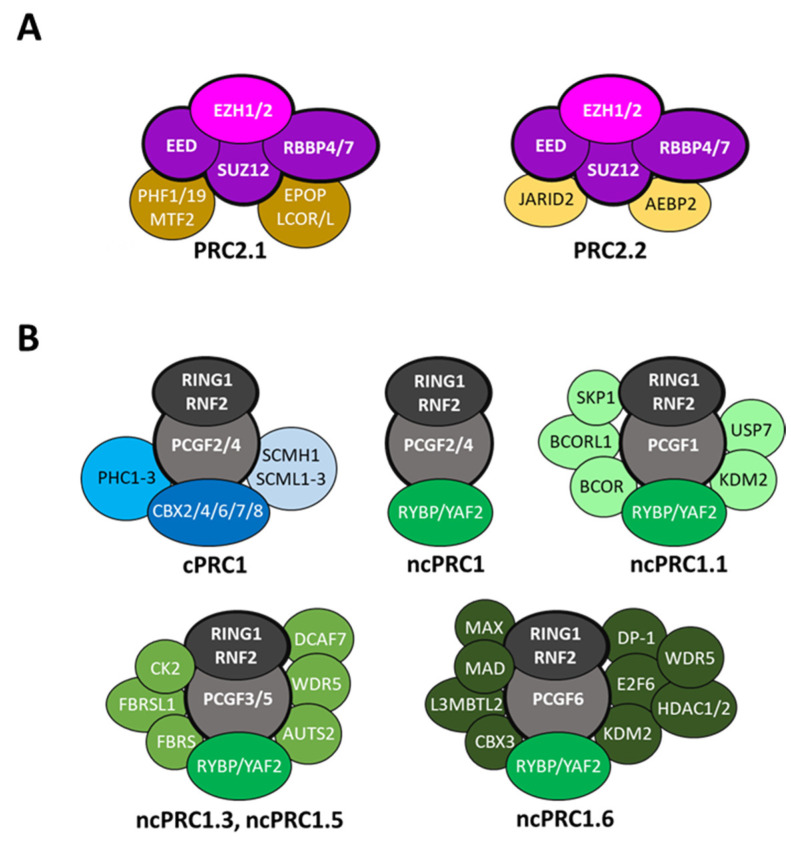
The mammalian Polycomb repressive complexes. (**A**) PRC2 is composed of four core subunits, EZH1/2, EED, SUZ12 and RBBP4/7, and is classified as two distinct complexes, PRC2.1 and PRC2.2 based on the auxiliary proteins associated. (**B**) The core PRC1 complex is composed of RING1 or RNF2 associated with one of the six PCGF proteins. Canonical PRC1 (cPRC1) includes PCGF2 or BMI1 (PCGF4), one of the chromobox proteins (CBX2, CBX4, CBX6-8), a polyhomeotic protein (PHC1/2/3) and eventually an SCM protein (SCMH1 or SCML1/2/3). In contrast, non-canonical PRC1 (ncPRC1) can include all PCGF proteins (PCGF1/2/3/4/5/6) associated with RYBP or YAF2. The identity of the PCGF subunit imposes the nature of the auxiliary proteins present in the assembly, thus defining diverse distinct ncPRC1 complexes.

**Figure 2 ijms-24-02322-f002:**
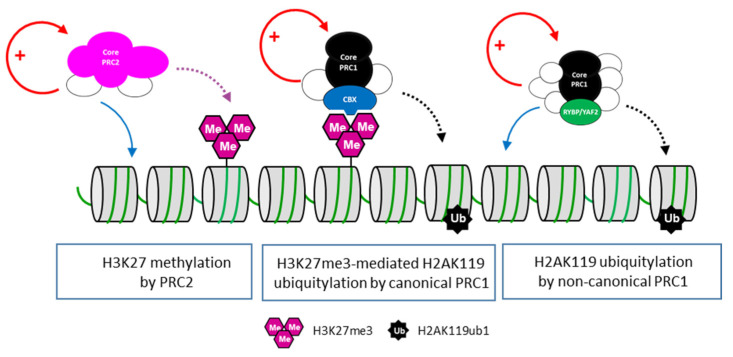
Deposition of the Polycomb-associated histone marks. Targeting of the PRC2 complex through the action of auxiliary proteins (white circles) leads to methylation at H3K27 (purple dashed arrow). This post-translational modification is recognized by the CBX subunit (blue) of the cPRC1 complex, which in turn ubiquitinylates H2AK119 (black dashed arrow). The ncPRC1 complexes containing RYBP/YAF2 (green) are recruited to chromatin, irrespective of H3K27me3, by the means of auxiliary subunits (blue arrow). Within all PRC complexes, some auxiliary proteins are involved in the stimulation of the catalytic activity (red arrows).

**Table 1 ijms-24-02322-t001:** Zebrafish genes coding for the PRC2 and PRC1 subunits.

Complex	Complex Type	Subunit	Zebrafish Ortholog
PRC2	Core	EZH2 EZH1 EED SUZ12 RBBP4 RBBP7	Ezh2 Ezh1 Eed Suz12a, Suz12b Rbbp4 - ^1^
PRC2	PRC2.1	EPOP PAL1 (LCOR) PAL2 (LCORL) PHF1 MTF2 PH19	Skida1l ^2^ Lcor Lcorl Phf1 Mtf2 Phf19
PRC2	PRC2.2	AEBP2 JARID2	Aebp2 Jarid2a, Jarid2b
PRC1	Core	PCGF1 PCGF2 PCGF3 BMI1 (PCGF4) PCGF5 PCGF6 RNF2 RING1	Pcgf1 - - Bmi1a, Bmi1b Pcgf5a, Pcgf5b Pcgf6 Rnf2 -
PRC1	cPRC1	CBX2 CBX4 CBX6 CBX7 CBX8 PHC1 PHC2 PHC3	Cbx2 Cbx4 Cbx6a, Cbx6b Cbx7a, Cbx7b Cbx8a, Cbx8b Phc1 Phc2a, Phc2b Phc3
PRC1	ncPRC1	RYBP YAF2 BCOR BCORL1 KDM2 USP7 SKP1 AUTS2 FBRS FBRSL1 E2F6 CBX3 HDAC1 HDAC2 L3MBTL2 WDR5 MAX MGA	Rybpa, Rybpb Yaf2 Bcor Bcol1 Kdm2ba, Kdm2bb Usp7 Skp1 Auts2a, Auts2b Fbrs Fbrsl1 E2f6 Cbx3a, Cbx3b Hdac1 - L3mbtl2 Wdr5 Max Mgaa, Mgab

^1^ Indicates that the zebrafish genome does not contain an ortholog. ^2^ *SKIDA1* and *EPOP* are derived from a common ancestral gene that was duplicated in early vertebrate evolution, giving the related gene *SKIDA1L*. In zebrafish, both *skida1* and *skida1l* are present in the genome, while in mammals, *EPOP* is found but *SKIDA1L* does not exist [24]. This suggests that *skida1l* could be orthologous to *EPOP*.

**Table 2 ijms-24-02322-t002:** The genes encoding PRC1 core subunits in the zebrafish (*Danio rerio*), the medaka (*Oryzia latipes*) and the Japanese pufferfish (*Tetraodon nigroviridis*).

Subunit	Zebrafish	Medaka	Japanese Pufferfish
RING1 RNF2 PCGF1 PCGF2 PCGF3 BMI1 (PCGF4) PCGF5 PCGF6	− + + − − ++ ++ +	+ + − + + + + +	− + + + − + + +

(−) Indicates that the gene is not found in the genome; (+) indicates that the gene is present in the genome as a singleton; (++) indicates that the corresponding subunit is encoded by two genes.

**Table 3 ijms-24-02322-t003:** The genes coding for the CBX subunits of the cPRC1 complex in zebrafish (*Danio rerio*), medaka (*Oryzia latipes*) and Japanese pufferfish (*Tetraodon nigroviridis*).

Subunit	Zebrafish	Medaka	Japanese Pufferfish
CBX2 CBX4 CBX6 CBX7 CBX8	+ + ++ ++ ++	++ ++ + ++ ++	++ ++ + ++ ++

(+) Indicates that the gene is present in the genome as a singleton; (++) indicates that the corresponding subunit is encoded by two genes.

**Table 4 ijms-24-02322-t004:** Genetic studies of zebrafish genes coding for PRC1 and PRC2 components.

Gene	Technique	Allele	References
*ezh1*	MO TALEN CRISPR/Cas9	- ul3 b1394	[68] [69] [70]
*ezh2*	MO ENU TALEN ENU CRISPR/Cas9	- hu5670 ul2 sa1199 ^2^ b1392	[71] [72] [73] [70,71,74] [70]
*eed*	TALEN	ul4	[75]
*suz12a/b*	CRISPR/Cas9		[76]
*rbbp4*	MO CRISPR/Cas9	- is60	[77] [78]
*rnf2*	MO ZFN CRISPR/Cas9	- ibl31 f5, f8	[79] [80] [81]
*pcgf1*	MO TALEN	- ul1	[82] [83]
*bmi1a/b*	MO	-	[84,85]
*phc1*	MO	-	[86]
*phc2a*	MO	-	[87]
*yaf2*	MO	-	[88]
*bcor*	MO	-	[89,90]
*kdm2bb*	MO	-	[91]
*usp7*	MO	-	[92]
*skp1*	IM ^1^	hi3970Tg	[93]
*wdr5*	CRISPR/Cas9	zju131	[94]
*auts2a*	TALEN	ncb104	[95]
*mgaa*	MO TALEN	- ihb801	[96] [97]
*max*	MO	-	[98]
*hdac1*	MO IM	- hi1618Tg	[99] [99]

^1^ IM, insertional mutagenesis based on the random integration of mouse retroviral vectors [100]; ^2^ ezh2 (sa1199) is a hypomorphic allele capable of supporting near-normal development [70,74].

## Data Availability

All relevant data are within the manuscript.

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
