# Peer review of "The Contribution of the Zebrafish Model to the Understanding of Polycomb Repression in Vertebrates"

_ijms, 2023, doi:10.3390/ijms24032322_

Round 1
Reviewer 1 Report
This work has appropriate and adequate references to related and previous work, and also use correct and readable English. It effectively helps us understand the relationship and the research progress between the Polycomb repression and the zebrafish mode.You can enhance the discussion of PRC1 and PRC2 functions in cancer, stem cell development, nervous system development, etc. I don't have much advice.
Author Response
We thank the Reviewer for its careful reading of the manuscript and its positive comments.
Reviewer 2 Report
In General, the authors provide a great overview of Polycomb Repression in Vertebrates. Nevertheless, I have a view points I would like to suggest for the authors to consider:
Mayor points:
In line 25, the authors quote a review, I would suggest to also give the original authors the credit they deserve for their findings.
Lines 87-89: I’m sorry but I don’t understand the sentence, could you please try and rephrase?
Lines 237-240: I would ask the authors to enlarge the paragraph about pharmacological approaches. Two sentences are very short.
Lines 286-289: Is this only challenging in zebrafish?
Lines 399-405: I would ask the authors to check the literature again. Especially about the role of nkx2.5 and what was shown in San et al., 2016
Minor:
Fig. 1: For me it was difficult to distinguish based on the chosen colors between core and auxiliary subunits in Fig. 1 A. In Fig. 1 B it’s not a problem at all.
Lines 209-212: In my understanding the current view in the field is, that MO experiments are accepted, the way it is phrased here sounds like the opposite.
Line 249: a little Typo: is it maybe supposed to be “has” or “have” not “as”?
I’d asked the authors to check when fish and when fishes is the right plural to use. In my understanding fishes is usually only used as plural for fish when more than one species is discussed.
Author Response
In General, the authors provide a great overview of Polycomb Repression in Vertebrates.
Response: We thank the Reviewer for its careful reading of the manuscript and its positive and constructive comments.
Nevertheless, I have a view points I would like to suggest for the authors to consider:
Mayor points:
In line 25, the authors quote a review, I would suggest to also give the original authors the credit they deserve for their findings.
Response: References corresponding to the pioneer works showing the recruitment of PcG proteins at numerous targets and the regulation of various target genes by the Polycomb repression is included. (Ref. 4, Boyer et al., 2006; Ref. 5, Bracken et al., 2006; Ref. 6, Tolhuis et al., 2006).
Lines 87-89: I’m sorry but I don’t understand the sentence, could you please try and rephrase?
Response: The sentence has been rephrased as “Thus, a fundamental difference between cPRC1 and ncPRC1 relies on whether their recruitment to chromatin depends on PRC2-deposited H3K27me3. cPRC1 complexes are recruited to H3K27me3 via the recognition of the marks by the CBX2, 4, 6-7 proteins, whereas ncPRC1 assemblies are recruited at chromatin by their auxiliary subunits, independently of H3K27me3” for a better understanding.
Lines 237-240: I would ask the authors to enlarge the paragraph about pharmacological approaches. Two sentences are very short.
Response: We thank the Reviewer for this comment and the paragraph on pharmalogical approaches has been enhanced.
Lines 286-289: Is this only challenging in zebrafish?
Response: No, that’s right. The sentence has been modified.
Lines 399-405: I would ask the authors to check the literature again. Especially about the role of nkx2.5 and what was shown in San et al., 2016
Response: We are grateful to the Reviewer for raising this important point. The paragraph has been modified.
Minor:
Fig. 1: For me it was difficult to distinguish based on the chosen colors between core and auxiliary subunits in Fig. 1 A. In Fig. 1 B it’s not a problem at all.
Response: The colors of the PRC2 auxiliary subunits has been changed.
Lines 209-212: In my understanding the current view in the field is, that MO experiments are accepted, the way it is phrased here sounds like the opposite.
Response: We made changes to reinforce the role of the MO-based approach.
Line 249: a little Typo: is it maybe supposed to be “has” or “have” not “as”?
Response: Thank you.
I’d asked the authors to check when fish and when fishes is the right plural to use. In my understanding fishes is usually only used as plural for fish when more than one species is discussed.
Response: Thank you, we went through the manuscript to correct when required.